# A novel task to investigate vibrotactile detection in mice

**Mariel Muller[1,2], Cyriel M. A. Pennartz[1,2], Conrado A. Bosman** [1,2]*, **Umberto Olcese[1,2]***

**1** Cognitive and Systems Neuroscience Group, Swammerdam Institute for Life Sciences, University of Amsterdam, Amsterdam, The Netherlands, **2** Amsterdam Brain and Cognition, University of Amsterdam, Amsterdam, The Netherlands

\* c.a.bosmanvittini@uva.nl (CAB); u.olcese@uva.nl (UO)

## Abstract

Throughout the last decades, understanding the neural mechanisms of sensory processing has been a key objective for neuroscientists. Many studies focused on uncovering the microcircuit-level architecture of somatosensation using the rodent whisker system as a model. Although these studies have significantly advanced our understanding of tactile processing, the question remains to what extent the whisker system can provide results translatable to the human somatosensory system. To address this, we developed a restrained vibrotactile detection task involving the limb system in mice. A vibrotactile stimulus was delivered to the hindlimb of head-fixed mice, who were trained to perform a Go/No-go detection task. Mice were able to learn this task with satisfactory performance and with reasonably short training times. In addition, the task we developed is versatile, as it can be combined with diverse neuroscience methods. Thus, this study introduces a novel task to study the neuron-level mechanisms of tactile processing in a system other than the more commonly studied whisker system.

**Data Availability Statement:** All Data and code used in this paper are available from https://gitlab.com/csnlab/olcese-lab/vibrotactile_detection_in_mice.

## Introduction

Touch is an essential sensory modality and understanding how somatosensory information is processed within the central nervous system is a long-standing goal for neuroscientists [1–4]. Although touch has been studied across various species, in the last two decades mice have emerged as the prime animal model to investigate microcircuit-level mechanisms of touch perception, plasticity, and related sensory-motor transformations [3, 5–14]. Mice have the advantage of enabling to combine the application of techniques able to record and manipulate the various components of the cortical microcircuit with behavioural tasks, using an adequate control of the delivery of sensory stimuli [15]. In addition, techniques such as transgenic manipulations, optogenetics, large-scale electrophysiology, and two-photon calcium imaging have vastly expanded our understanding of how somatosensory information is processed and transformed into motor responses by enabling us to probe the functionality of all components of the cortical and subcortical microcircuitry [16, 17].

**Funding:** This study was supported by this work was supported by the European Union's Horizon 2020 Framework Program for Research and Innovation under the Specific Grant Agreement 785907 (Human Brain Project SGA2) and 945539 (Human Brain Project SGA3) to CAB, UO and CMAP, by the FLAG-ERA JTC 2019 project DOMINO (co-financed by NWO) to CAB and UO, by an Amsterdam Neuroscience Proof of Concept grant to UO and by intramural funds from the Swammerdam Institute for Life Sciences of the University of Amsterdam to UO, CAB and CMAP. The funders had no role in study design, data collection and analysis, decision to publish, or preparation of the manuscript. There was no additional external funding received for this study.

**Competing interests:** The authors have declared that no competing interests exist.

Most of the knowledge we have about the neuronal basis of somatosensation was acquired via studies focused on the whisker/barrel system in rodents, which has long been used as a model for human somatosensory processing [5, 6, 18, 19]. In rodents, the barrel field occupies a large fraction of the somatosensory cortex, and, thanks to the highly specific mapping between individual whiskers and topographically organized groups of neurons, this model system has allowed to understand many properties of tactile processing, ranging from stimulus selectivity [6, 20] to experience-dependent plasticity [21–23]. Nevertheless, interpreting and translating the results from the whisker system to humans can be challenging [18, 24]. There are significant functional and morphological differences between the whisker system and touch processing in humans [18]. To begin with, mice use their whiskers to navigate and explore their peripersonal space, while humans use their hands to grasp and manipulate objects, but navigate mainly with vision [25]. Moreover, humans explore objects with their hands, and sensory signals are shaped by many contact points on the skin's surface [26, 27]. On the other hand, mice explore objects sparsely, and incoming sensory signals are mainly determined by forces applied to their whiskers [25]. These differences result in distinct sensory representations [6], which make direct translation difficult. Furthermore, the microcircuit-level organization of the barrel system, both at the thalamic and cortical level, shows major structural differences from other skin-related somatosensory areas. For instance, the barrel field of the primary somatosensory cortex is topographically organized in a series of patches primarily containing neurons responding to stimulation and motion of individual whiskers; these patches are surrounded by septal areas, not specifically tuned to any whisker [6, 20, 28]. In contrast, the skin-related primary somatosensory cortex shows a more homogeneous surface structure and mainly presents neuronal responses to touch and not motion [29–32]. These differences highlight the importance of developing a method to probe tactile processing beyond the mouse whisker system.

Surprisingly, very few tasks have been developed to probe tactile perception and decision-making in mice in a way that engages skin-related somatosensation and not the whisker system. The only task that, to our knowledge, addresses this issue is a Go/No-go task in which mice learned to discriminate the frequency of an applied vibration [33, 34]. Consequently, our understanding of the microcircuit-level physiology of somatosensation in—for instance—limbs and paws mostly comes from studies involving passive stimulation of mechanoreceptors [35, 36]. The limited availability of tasks suitable to study skin somatosensory perception is problematic, since engagement in a task has been shown to strongly influence how sensory information is processed within the thalamocortical system [11, 15, 37, 38]. Thus, passive stimulation of receptors is insufficient to fully understand how somatosensory processing operates.

To address this, we developed a novel vibrotactile detection task for head-fixed mice that provides an alternative framework to those available for the barrel/whisker system to study the detection of tactile stimuli as a function of the intensity of the applied stimulus [18]. This type of task is extensively used in the study of other sensory modalities such as vision and audition, but is strikingly not available for limb somatosensation. In this task, a vibratory stimulus is delivered to the mouse's hindlimb, such that mechanoreceptors in the mouse skin are stimulated similarly to what typically occurs in humans [27, 27, 39]. Importantly, detection performance scaled with vibration intensity, and mice learnt to ignore the sound produced by vibratory motors. By showing that mice can successfully be trained to perform this task, we have provided an alternative tool that neuroscientists can use to investigate the neuron-level mechanisms of touch perception and decision-making in a system other than the barrel/whisker cortex.

## Methods and results

In this section, we combine methods and results to provide a comprehensive overview of the task design, the training procedure, and the results obtained with the task we designed.

### Subjects

A total of 4 adult, male transgenic Pvalb-IRES-Cre (B6;129P2-Pvalb<tm1(cre)Arbr>/J) mice *(Mus musculus)*, three months old, were used as subjects in this study. This transgenic line is often used in studies in which area-specific optogenetic inactivation is used, and has been successfully used as an experimental model to study sensory processing and decision making—see e.g. [8, 37, 40]. Comparable results are therefore expected with different mouse lines. Mice were bred in homozygous colonies at the Animal Facility of the University of Amsterdam, and housed in small groups of siblings. They were kept on a 12 h reverse day-night cycle (lights off at 08:00hs and on at 20:00hs) with controlled values of humidity (55% ±10) and room temperature (21.5 ± 2.0 ˚C). Before behavioural training, water was provided *ad libitum*. During training, a gradual water restriction protocol was followed (see below). Food was always provided *ad libitum*. After completion of behavioural procedures, mice were euthanized. All procedures were approved by the Dutch Commission for Animal Experiments (Centrale Commissie Dierproeven) and by the Animal Welfare Body of the University of Amsterdam, under permit AVD1110020172385. Written consent for all procedures has been obtained.

### Surgical procedures

Mice were implanted with a light-weight titanium head bar aiming to headfix them in the set-up. All mice were anesthetized with Isoflurane (IsoFlo, 250ml), with a 3% concentration for induction and 1.5% for maintenance. Carprofen (5mg/kg, injected s.c.) was administered as analgesia. Breathing rhythm and rectal temperature were monitored during the whole procedure. We protected the eyes with an ophthalmic ointment (Ophtosan, ASTfarma). We removed scalp hair with a hair removal cream (Veet). A local anaesthetic (Xylocaine 10%) and iodopovidone (antiseptic, Betadine, 30ml, Mylan) were applied to the skin over the scalp. After a linear skin incision, the scalp was cleared, dried, softly scratched to facilitate adherence to the head bar and covered by a layer of cyanoacrylate glue (Loctite 458). The temporal muscle was slightly retracted. We placed the head bar on the skull in a region encompassing lambda and bregma sutures and parietal bones. The final attachment was done with dental cement (shade 528/1 pink, Kemdent) followed by dental acrylic (Superbond C&B). After this surgery, mice had one week for recovery with water and food *ad libitum*.

### Behavioural set-up

Mice were trained in a custom-made set-up constituted by a metal platform for head fixation, a lick spout, an infrared sensor (IR), two vibratory motors, and a Polyvinyl chloride (PVC) tube (Fig 1A, S1A Fig). The PVC tube provided body stabilization and shelter. Mice spontaneously entered the PVC tube, and, when they extended their heads outside the tube, the head bar was screwed onto a head fixation support. In this way, animals remained oriented towards the infrared sensor monitoring licks and the lick spout.

Before starting a behavioural session, the right hindlimb was gently retracted and surrounded by a flexible plastic ring. Following this, the paw was fixed to a soft foam earplug. The plastic ring was attached to a vibratory motor (LilyPad- SparkFun-DEV11008, where the motor is a 310–101 10mm shaftless vibration motor from Precision Microdrive) attached to a pole placed on the right side of the metal platform. Close to this motor, we placed a second

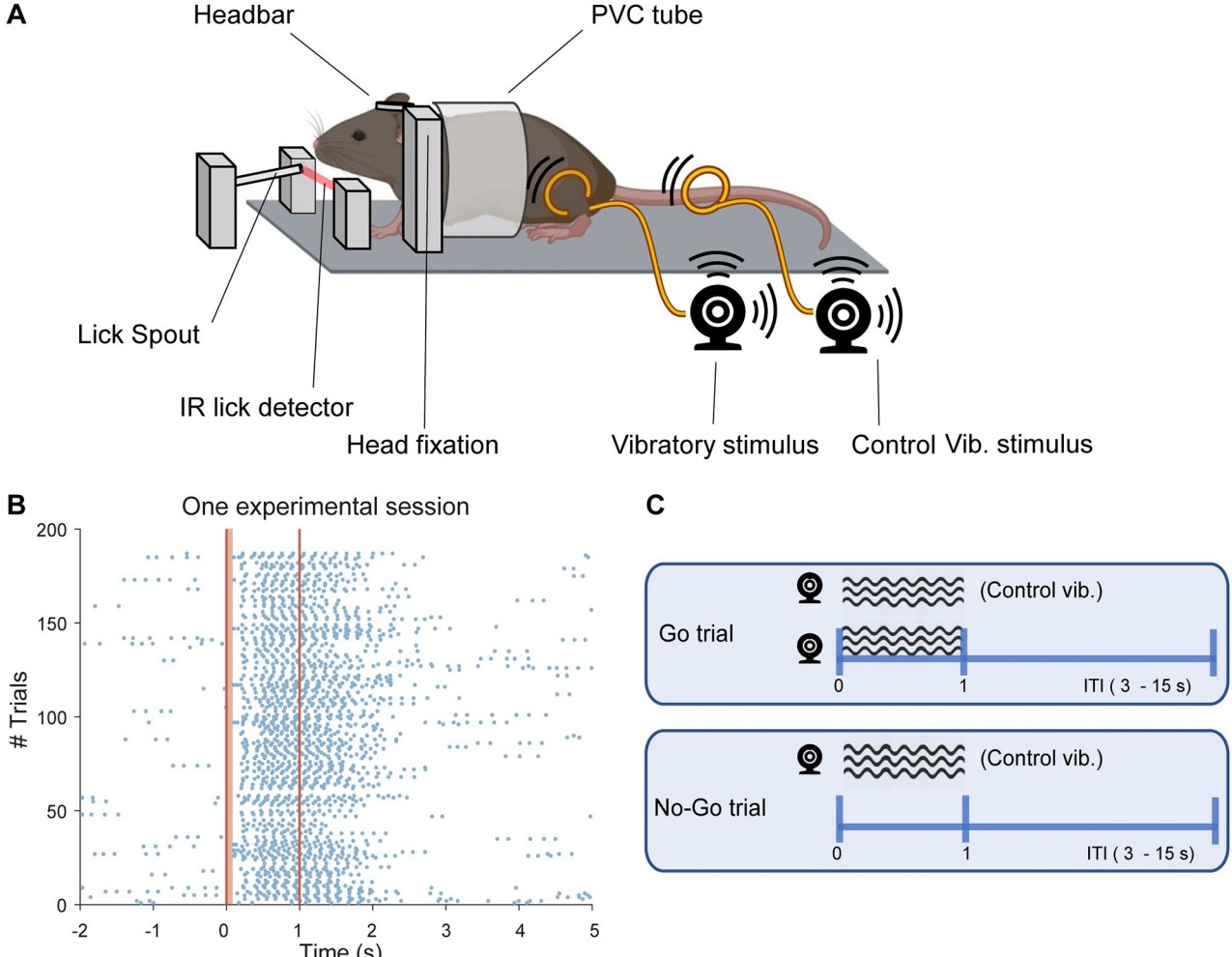

**Fig 1. Vibro-tactile behavioural task.** (A) Schematic representation of the behavioural set-up. The mouse enters the PVC tube, it is fixed to the head-fixation supports, and its hindlimb is surrounded with a plastic cable (orange) which delivers a vibrotactile stimulus from an attached vibratory motor. A second motor, not attached to a limb, is used to deliver a control vibration during probe trials and verify that mice do not respond to the sound produced by vibratory motors. A lick spout delivers a reward if a lick is detected by an infrared (IR) lick detector following the delivery of tactile stimuli. Reprinted from Biorender under a CC BY license, with permission from Biorender, original copyright 2023. See also S1A Fig for a photo of the setup. (B) Lick response raster plot from one session recorded during the Final Stage. Blue dots correspond to individual licks. The brown bar between 0 and 0.1 s indicates a no reward window. Licks detected in this window are not rewarded, accounting for physiological reactions time. Dark vertical lines at 0 and 1 second indicate the response window. Note that the lick distribution is time-locked to stimulus presentation, indicating that the mouse responded to tactile stimuli. (C) Outline of the Go and No-Go trials (Final Stage). Black circles and waves represent the vibratory motors and their stimuli. Stimulus time goes on from 0 to 1 second and ITI lasts from 3 to 15 seconds, randomly. A reward is only delivered upon licking during Go trials. Panel A was created with BioRender.com. *Vib.: vibratory; ITI: intertrial interval.*

motor not attached to the animal, that we used as a control for the noise produced by the first vibratory motor (see later). The proximity of the two motors (S1B Fig) makes it unlikely that mice could distinguish them based on auditory features. In particular, the angle between the two motors was set at about 10 deg (see S1B Fig). This is lower than the minimal auditory angle of 31 deg that has been reported in mice [41].

Every time the tongue of a mouse reached the lick spout it crossed the light beam emitted by the IR sensor. In this way licks were detected as binary events by an Arduino board (Mega-2560) and sent to a PC. A reward could be delivered via a small tube ending on the lick spout. The amount of reward was controlled via a valve (Biochem Valves—075P2NC24-02SQ).

## Delivery of sensory stimuli

An Arduino board (Mega 2560) was used to control the intensity and duration of the vibration. Intensity could be modulated on a 0–255 pulse width modulation (PWM) scale level. In the training stages, PWM scale value was set at the maximum value, corresponding to an intensity of 3.6–3.8 G with a vibration frequency of 210–220 Hz.

Since vibratory motors also produced a sound, we controlled for the possibility that mice could learn to respond to the sound and not to the vibration by employing a second motor that was not attached to the animal (Fig 1A, S1 Fig). This motor was used to produce a sound during catch (probe) trials, so that mice would learn not to associate the sound produced by motors to a reward.

## Statistical analysis

All data are expressed in mean and ±standard deviation (SD) unless otherwise noted. All analyses were performed with MATLAB (The Mathworks) software and custom-made scripts. Details about the statistical analyses are presented in the following sections.

All Data and code used in this paper are available from https://gitlab.com/csnlab/olcese-lab/vibrotactile_detection_in_mice.

## Training stages and results

**Initial habituation.** Different odours, tail handling and changing of experimental rooms are some of the factors that can stress mice [9, 42, 43]. Therefore, mice first had to be habituated to the experimenters. For this purpose, the experimenters initially placed their hands on the home cage and waited until the mice felt comfortable enough to approach it and walk over it. This procedure took 2–3 days. In addition, plastic tubes were placed in their home cages, and they were used to transport mice from inside to the outside of their home cages once mice felt comfortable enough to enter the tubes.

**Controlled water uptake.** All mice were placed under a water rationing protocol before training. This was necessary to motivate animals to perform the task [9, 44–48]. The protocol consisted in restricting their access to water gradually over several days. Normally, *ad libitum* water intake is between 3 to 6 ml/day [48]. A transition from *ad libitum* to rationed intake was done over the course of 3 days, providing 3 ml/day on the first day, 2 ml/day on the second day, and 1 ml/day on the third day. The welfare state, weight curves and dehydration signs were monitored daily.

Training involved one behavioural session per day, between 13:00 to 16:00, from Monday to Friday. During the session, each mouse was expected to obtain between 0.7 to 2.1 ml of reward (infant formula, 0–6 months, Kruidvat) per day. If they obtained less than this range of reward volume, a supplement of water was offered. During weekends, there were no training sessions, and mice received water in a gel form (HydroGel™). A minimum of 0,045 mg/g/day of gel was administered. If the weight of a mouse fell below its 85% pre-restriction weight or there were signs of dehydration or pain, the water restriction was stopped, and mice had free access to water. During this period, the animal was set aside from the training session to recover weight and welfare signs. Only when the mouse recuperated, water restriction protocol could be applied again.

**Habituation to the set-up.** Following the initial habituation, there was another period of habituation during which the mouse was set over the metal platform that upholds the PVC tube (Fig 1A, see also S1A Fig) and left to explore it for 15–25 min. After 2–3 days of exploration, we placed the mouse in one end of the PVC tube, and the mouse could receive a small reward when it reached the opposite side. On that side of the tube, a lick spout was present

which delivered drops of infant formula. When the mouse was comfortable with reaching the lick spout, the head bar's wings were held over the metal support and screwed. The mouse initially remained head-fixed for about 15 min. The same procedure was then repeated on the next days while gradually increasing the duration of the fixation period in steps of 15 minutes, until reaching 1 hour. Habituation to the head-fixation on average took 8 sessions in total. Next, habituation of the paw fixation was implemented within a few days.

**Training Stage 1.** Following habituation, a passive stage was implemented (Stage 1). The goal of this stage was to teach animals to associate vibrotactile stimuli to the delivery of a reward. The mouse was head and paw-fixed, and each trial was composed of a stimulus period and an intertrial interval (ITI). During the stimulus period a 2 s-long vibration at maximal intensity was applied to the hindlimb. This was coupled with an automatic reward (6 µl of infant formula). Intertrial times were fixed to 20 s (Fig 2A—Stage 1). Assessment of performance was based on the amount of milk consumed at the end of each session (a range of 1–2 ml/session) and we verified the absence of spill-over around the lick spout. Of relevance, reaction times could potentially also be used to monitor mouse behaviour during Stage 1. All mice learnt to lick to obtain rewards within a period of 4 days, but we decided to extend Stage 1 for up to 13 days (Fig 2B) before proceeding to Stage 2. We chose this longer habituation period because we wanted to make sure that mice were well habituated to the procedure of positioning the vibratory ring to the hindlimb. Furthermore, experimenters also needed some sessions to learn to master this step, in order to minimize discomfort to the mice and ensure a stable positioning. Stage 1 could therefore be shortened in future experiments. Each session had a duration of 45–60 minutes and the trials per session were not counted.

**Training Stage 2.** Following Stage 1, all subsequent training stages were active, i.e., reward was only delivered when mice responded to the presentation of a vibratory stimulus by licking towards the lick spout. Thus, after associating stimulus presentation to reward delivery (Stage 1), in Stage 2 mice had to learn to perform licking actions to obtain a reward, and to time licking actions to sensory stimuli. Each trial consisted of a stimulus period with a vibration lasting for 2 s. Intertrial periods had a random duration between 5 to 20 s based on an exponential distribution wit (Fig 2A, Stage 2) [37]. The exponential distribution had a mean of 1 s, with a minimum (offset) value of 5 s and a cut-off at 20 s. Note that the probability of having an ITI longer than 20 s is smaller than 0.001%. Thus, the hazard rate can be assumed as flat for ITI values larger than 5 s. Each trial was categorized as a 'Hit' when a lick was performed during the stimulus period and after a grace period of 100 ms from stimulus onset, to account for physiological reaction time [37]. A reward was delivered immediately following the detection of the first lick. 'False alarms' (FAs) responses were estimated by considering licks detected during ITIs, in the one second window before the stimulus period (a window that was chosen to assess spontaneous lick rate in the vicinity of stimulus delivery and that has temporal properties comparable to those of regular trials). The FA rate was computed as the fraction of 1 s windows preceding stimulus onset that included at least one lick event, which were thus considered as surrogate FAs, or trials in which mice would have licked even in the absence of sensory stimuli [37].

Performance was measured by computing the *D-prime (d')* sensitivity index [15, 49], that measures the difference between Hit rate and FA rate. D-prime was computed via the following formula:

$$d' = \phi^{-1}\left(Hit\ rate\right) - \phi^{-1}\left(FA\ rate\right) \tag{1}$$

where $\phi^{-1}$ is the inverse of the cumulative normal distribution of the hit rate and the false

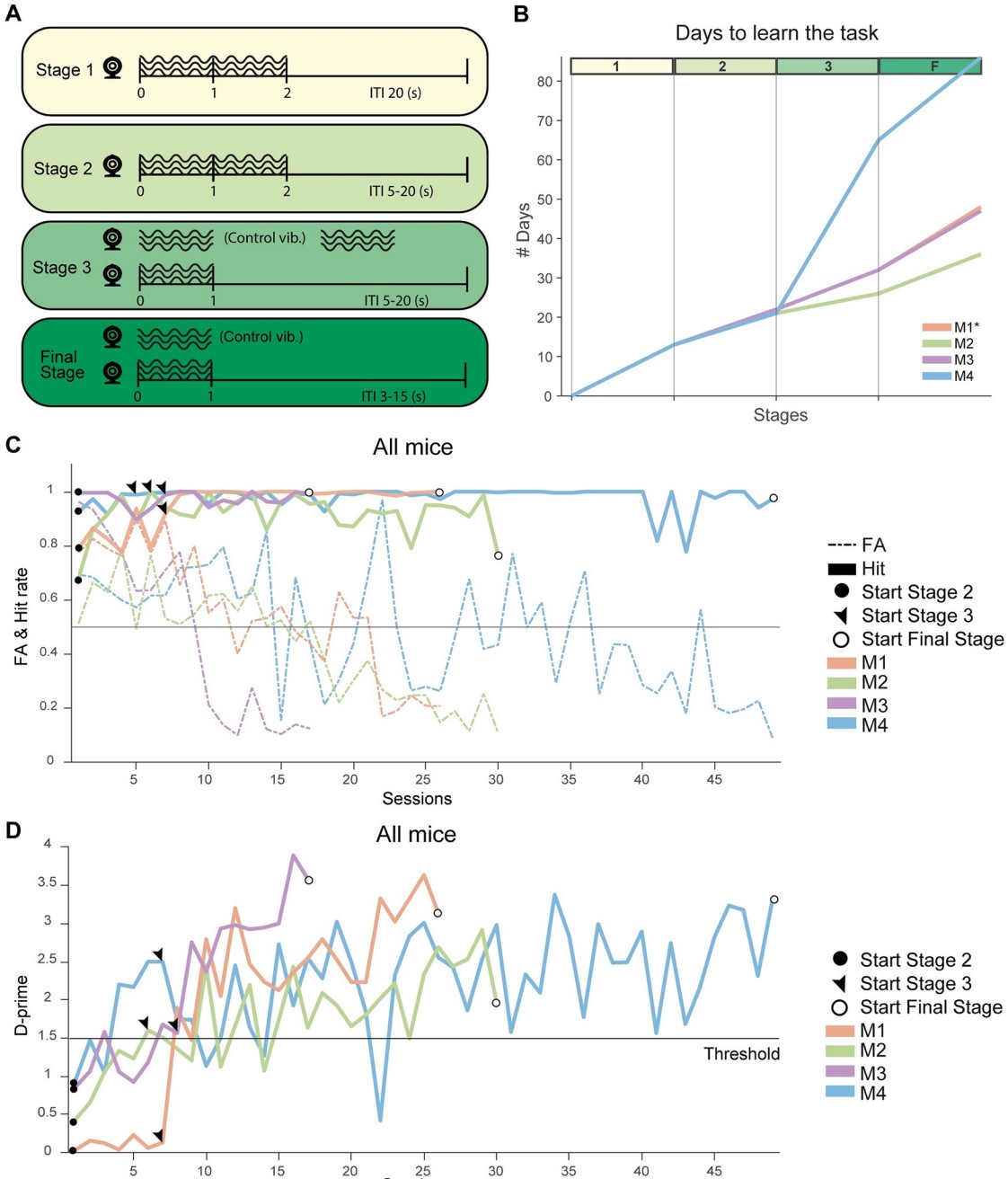

**Fig 2. Training stages and mice performance. (A)** Schematic representation of the training Stages. From top to bottom: Stages 1, 2, 3 and Final Stage (each characterized by a different colour, from yellow to dark green). The black circle and waves represent the vibratory motors and their stimuli. Stimulus time and ITIs are longer in the beginning and are reduced across stages. Control vibrations are introduced in Stage 3 and in the Final Stage. **(B)** Average Stage progression (measured in days) for all mice. The y axis shows the number of days. Coloured bars on the top indicate the corresponding training stage. Orange, Green, Violet and Blue lines are the learning progression curve of each mouse (M1-4). Values for M1 overlap with those of M3, thus its curve is hardly visible in the graph. **(C)** Time course of Hit and FA rates (solid and dashed lines, respectively) across Stage 2 and 3 all four mice (same colour coding as in panel *B*). The grey line at 50% defines the chance response level. This figure shows how FAs are decreasing across sessions, gradually, from 90% in the first session of Stage 2, to less than 25% at the end of Stage 3. Moreover, more than 85% of responses are Hit responses, a value that stays stable across sessions. **(D)** *D-prime* values across Stages 2 and 3 for all four mice (same colour coding as in panel *B*). There was a consistent increment in performance along Stage 2 and 3. The horizontal line indicates the D-prime of 1.5 that is commonly considered to be satisfactory performance. Note that both D-prime and FA rate were used to determine the transition from Stage 3 to the Final stage. *ITI*: *intertrial interval; FA*: *false alarm*.

alarm rate. *We adjusted values of Hit and FA rates that were equal to 1 or 0 to either 0.99 or 0.01*, respectively [50].

However, the D-prime index was not used to assess behavioural performance, as the goal of Stage 2 was to train animals to associate vibratory stimuli to the lick-related delivery of a reward. The absence of licks during the stimulus period was considered as a 'Miss' response and the absence of licks during the one second window before the stimulus period was considered as a Correct Rejection response (CR). Mice were not punished for False alarms or Miss trials.

In this stage, FA rates were still high (Fig 2C) and our criterion for going to the next stage was based on the hit rates, lick response raster plots and peristimulus time histograms (PSTHs). One of the criteria to go to Stage 3 was the presence of 4 or more sessions with hit rates higher than 80%. The other criterion was the presence of a significant peak in lick PSTHs during stimulus time, this peak is time locked to the beginning of the stimulus period. At the end of Stage 2, there had to be a clear increase of licking rate during stimulus presentation in comparison with licks happening before and after it. After 8–9 sessions (SD 0.5), all mice were ready for Stage 3 (Fig 2B).

**Training Stage 3.** In Stage 3, stimulus time was reduced from 2 s to 1 s and control vibrations were introduced. The goal of Stage 3 was to train animals to respond to vibrotactile stimuli while ignoring sounds produced by the motors. Each trial started with two vibrations at the same time, one delivering the tactile stimulus to the hindlimb of the animal and the other being a control vibration coming from the second motor (Fig 2A-Stage 3), not attached to the animal's paw. In the course of the intertrial period, we added another control vibration lasting 1 s, with a random onset but still respecting the minimum duration of the ITI (i.e., 5 s). This allowed us to control for false alarm rates, and to train mice not to respond to control vibrations. False alarm responses were based on licks happening during control vibrations within ITIs. The absence of licks during control vibrations presented within ITIs were considered as a CR response, their presence as a FA. Hit responses were licks detected during the stimulus period and their absence was considered a Miss response. After obtaining at least 4 consecutive sessions with *D-prime* over 1.5 and a FA rate lower than 0.25, mice were ready for the Final Stage.

One mouse (M2 –green line) finished Stage 3 in 5 sessions, two mice (M1—violet line, M3 –overlaps with M1) finished it in 9–10 sessions and the last one (M4 –blue line) finished it in 44 sessions (Fig 2B, one session per day, from day 20 to day 64). On average, mice took 17 days (SD 18) to complete this phase.

**Final stage.** In the Final Stage we introduced trials with tactile stimuli at different intensities: low, medium, and high. We also implemented probe trials with comparable timing features to regular trials but only a control vibration, and we no longer played control vibrations during ITIs. The goal of the Final Stage was to be able to obtain psychometric curves quantifying task performance.

A tactile started with 1 second of tactile vibration coupled to a control vibration. This was followed by an intertrial interval with a random duration of 3 to 15 seconds (exponential distribution, mean ITI duration of 4 s, Fig 1C). Licking during the tactile vibration was rewarded with 6 μl of baby milk formula and was considered a Hit response (Fig 1C, upper panel). When there was no licking activity during the tactile vibration, we counted this as a Miss trial.

A probe trial started with a control vibration followed by the aforementioned ITI (Fig 1C, lower panel). Licking activity during probe trials was not rewarded and it was categorized as FA response. When there was no licking activity during probe trials, we counted this as a CR.

Probe and tactile trials were randomly distributed within a session.

As mice learnt to respond to a tactile stimulus successfully in previous Stages, ITIs could be shortened without affecting their performance (Fig 2A, Final Stage). To ensure that mice did not associate auditory noise to reward, we always included 50% probe trials and 50% tactile trials (in total, encompassing low, medium, and high intensity). Tactile intensities were determined per animal, by using psychometric curves computed at the end of each session. Psychometric curves are obtained by fitting a cumulative gaussian function to the three data points corresponding to the proportion of Hits and the data point corresponding to FAs during each session [37, 51]. Specifically, we used three intensity ranges that matched to a below-chance response (<50%, low intensity: 1.8–2.25 G at a frequency of 105–130 Hz), a higher than chance but moderate response (50–80%, medium intensity: 2.6–2.9 G at a frequency of 150–175 Hz), or a higher than 80% hit rate (high intensity: 3.3–3.8 G at a frequency of 195–210 Hz). When the proportion of Hits matched the response expected as low, medium or high, we fixed these tactile intensities for the following sessions. Otherwise, we varied the tactile intensity to achieve suitable detection rates. Of relevance, mice did not need additional training sessions to habituate to the novel stimulus intensities. This suggests that the different vibratory frequency that the type of vibratory motors we used produce as a function of applied input voltage and vibratory intensity was either not perceived by mice, or did not affect their detection ability, which was rather mainly influenced by intensity.

**Task performance.**   Mice performed an average of 424 (SD 80.7) trials per session in stages 2 and 3, and an average of 484 (SD 147) trials per session at the Final Stage. The number of trials per sessions was not assessed during Stage 1. The required average time to progress from Stage 1 to Stage 3 was 39 (SD 17.7) days and after that reaching the final Stage required an average of 16 days (SD 4.5) (Fig 2B). Behaviour was evaluated via several measures: *D-prime* values, FA rates and parameters describing the psychometric curves.

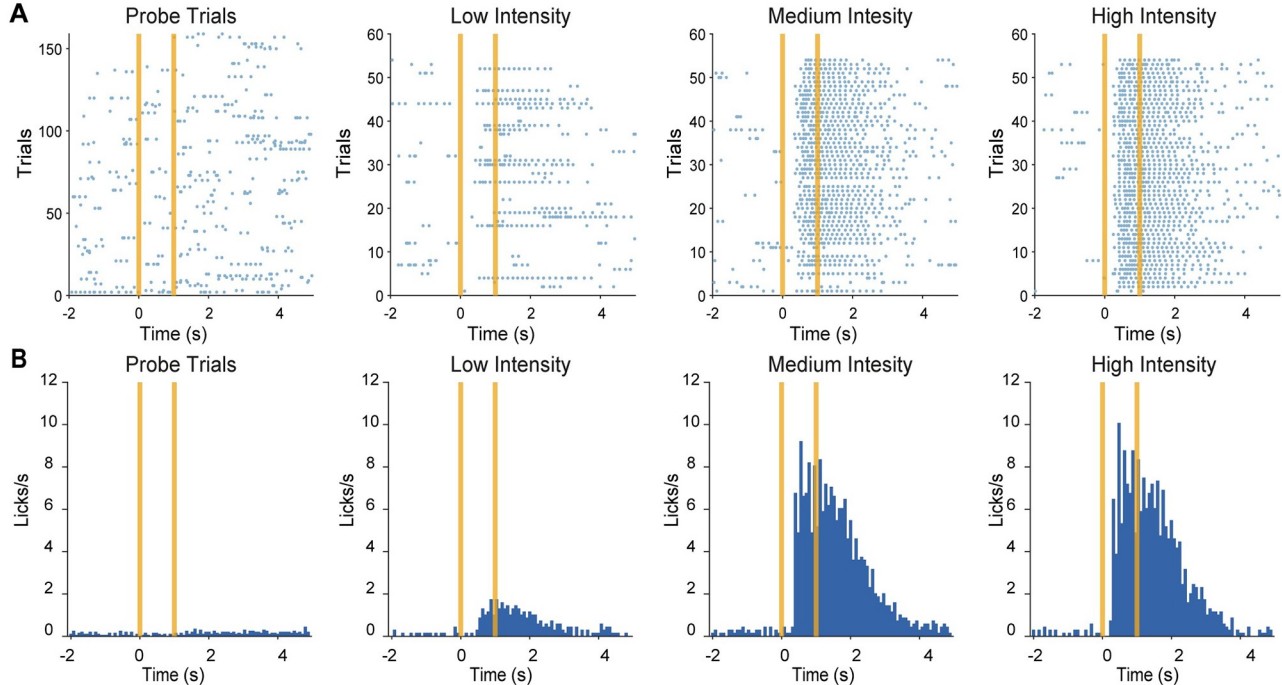

**Fig 3. Licking behaviour. (A)** Licking activity during one example session from M3 in the Final Stage (blue dot: individual lick). From left to right, each lick raster plot corresponds to increasing tactile intensities from probe trials to low, medium, and high intensity tactile trials. The vertical lines indicate the start and end of sensory stimuli. **(B)** Same as panel *A*, but for lick peristimulus time histograms (time bin: 80 ms).

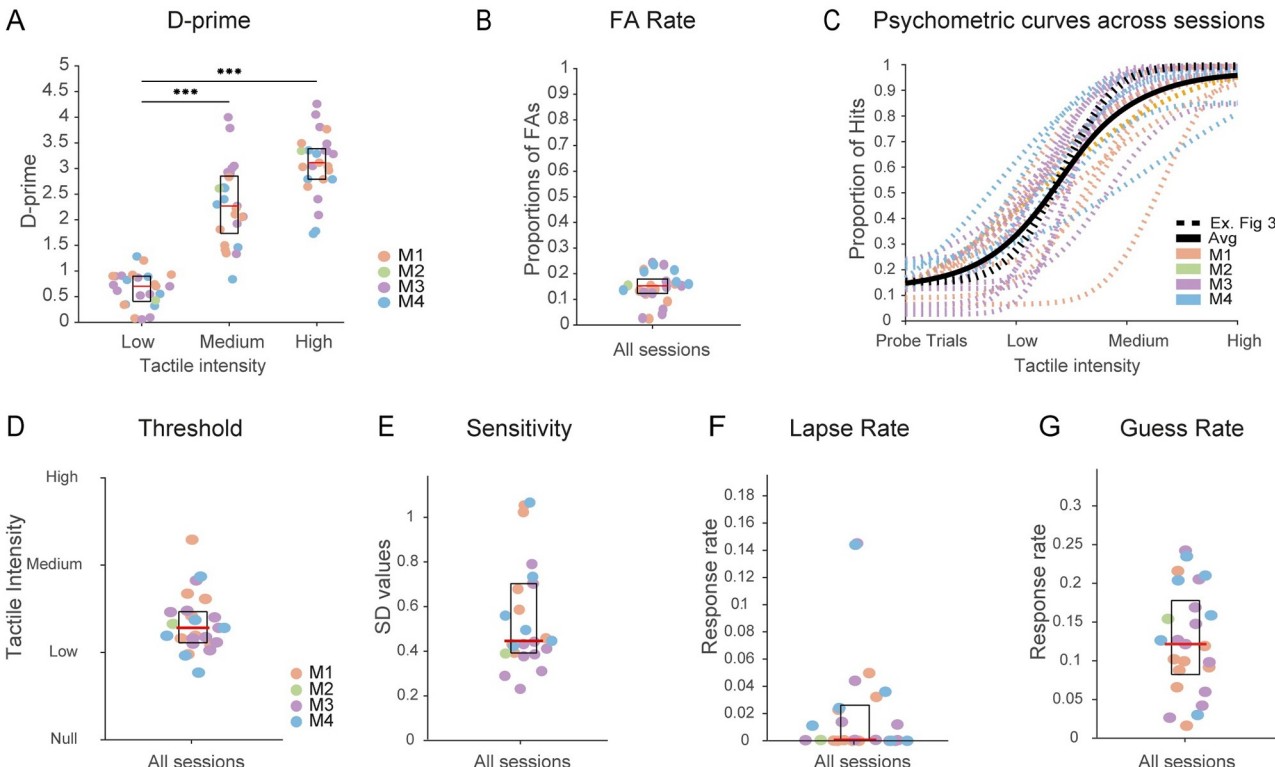

**Fig 4. Behavioural performance. (A)** *D-prime* values for each of the tactile intensities (Low, Medium, High). Orange, Green, Violet, and Blue dots correspond to each mouse (M1-4). ***Significant difference between low and medium tactile intensity (one-way ANOVA, $F(2,72) = [105.1]$, $P = 4.44$ x $10^{-22}$, Bonferroni post hoc test, $P = 4.7$ x $10^{-14}$) as well as between low and high intensity (one-way ANOVA, $F(2,72) = [105.1]$, $P = 4.44$ x $10^{-22}$, Bonferroni post hoc test, $P = 4.23$ x $10^{-22}$). **(B)** FA rates across all sessions in the Final Stage. **(C)** Psychometric curves of all mice. Coloured dashed lines are individual sessions belonging to each mouse (M1-4). The solid black line is the average from all sessions. The dashed black line corresponds to the example session shown in Fig 3A and 3B. **(D-G)** Threshold, sensitivity, lapse rate and guess rate computed from the psychometric curves. NS.: Statistical significance was not found between mice for any of these five features (one-way ANOVA).

An example of licking behaviour to the various conditions is shown in Fig 3. Fig 3A shows lick response raster plots and Fig 3B displays lick peri-stimulus time histograms (PSTHs) as a function of vibration intensity. A clear trend of increasing licking activity can be observed going from probe trials to the highest tactile intensity (High Intensity condition). During probe trials low lick rates can be observed, not clearly locked to stimulus presentation (Fig 3A and 3B, leftmost panels). Thus, mice did not respond to the sound produced by motors, since this was also present in probe trials. When a vibrotactile stimulus was instead delivered, lick responses became time locked to stimulus presentation. Lick rates also increased as a function of the intensity of the stimulus.

Fig 4A illustrates the *D-prime* values obtained per vibratory intensity across all sessions and mice. *D-prime* values along tactile conditions follows a clear gradual increase from low to high tactile intensity, confirming that the different intensity levels are accompanied by increasing detectability. Average *D-prime* responses and their standard deviations were 0.65 ±0.3 (mean ± SD), 2.29 ± 0.76 and 3.08 ±0.6, at low, medium, and high intensities, respectively. FA responses per session were below 25% of all probe trials (Fig 4B). The FA median response fraction was 15% (shown as a red line) and the average response fraction was 14.5%.

Next, we estimated the behavioural performance using the psychometric function [52]. Fig 4C depicts the resulting psychometric curves across animals. Each animal showed a low

response probability ($< 25\%$) for probe trials (reflecting FA rate) and a gradual increase towards a plateau—in most sessions—at the maximum intensity ($> 90\%$).

To compare differences between sessions, we analysed the psychometric results based on signal detection theory [15, 51, 53]. We computed the major features that can be extracted from psychometric curves (Fig 4D–4G): threshold, sensitivity, lapse rate, and guess rate.

To determine the threshold we measured the intensity at which the stimulus was detectable 50% of the time [15]. For this purpose, we took the intensity value corresponding to the 50% hit rate from each session. Fig 4D shows threshold values for each session and mouse. No significant differences were found between mice after one-way ANOVA test ($F(3,21) = [0.36]$, $P = 0.7$).

The psychometric curves presented in Fig 4C can also be characterized by sensitivity, which is shown in Fig 4E. Sensitivity refers to the slope of the psychometric curve [15, 51, 52]. It was computed as the standard deviation of the gaussian distribution of hit rate proportions [52], and we found no significant differences between mice (one-way ANOVA test ($F(3,21) = [1.95]$, $P = 1.15$)).

Next, the lapse rate is the probability of not responding to a stimulus, independently of stimulus intensity [52], and was computed as the difference between the highest possible performance (i.e., 1) and the upper bound of the psychometrics curve. Fig 4F shows the lapse rate values for each session and mouse. Average (not shown) and median (red line, 50% percentile) values were under 0.02%. There were no significant differences in lapse rates between mice (one-way ANOVA test ($F(3,21) = [0.43]$, $P = 0.73$)).

Finally, the lower bound of the psychometric curves corresponds to the guess rate. This quantifies the probability of making a mistake under absence of stimulation (i.e., a FA response). Fig 4G shows that the guess rate always remained under 25% (cf. Fig 4B). Differences between mice were not significant (one-way ANOVA test ($F(3,21) = [1.01]$, $P = 0.41$)).

Next, to evaluate task performance more in depth, we also measured the frequency of licks [54] in the response window and the latency of the first lick (also called reaction time) [7] across stimulus intensities. Fig 5A and 5B illustrate these results. Lick rates were estimated during the 1 s window of stimulus presentation (or the equivalent window during probe trials). We only included hit trials and FA trials. We found average values of lick response rate of 0.38 Hz (SD 0.16), 1.61 Hz (SD 0.8), 4.12 Hz (SD 1.13) and 5.25 Hz (SD 0.9) to probe trials, low, medium, and high intensities, respectively. There was a gradual and significant trend going from probe trials to high-intensity stimuli. Statistical analysis of lick rates was performed by repeated measures, one-way ANOVA with Bonferroni post hoc test. Probe trials were accompanied by lick rates significantly lower than all other tactile intensities (Fig 5A, one-way ANOVA, $F(3, 96) = [171.4]$, $P = 2.024 \times 10^{-38}$, Bonferroni post hoc test: probe vs. low intensity, $P = 1.09 \times 10^{-05}$; probe vs. medium intensity: $P = 4.35 \times 10^{-27}$, probe vs. high intensity, $P = 1.81 \times 10^{-35}$). As concerns the latency of the first lick in the response window (i.e. reaction time, Fig 5B), we found average values of 0.37 s (SD 0.06), 0.5 s (SD 0.4), 0.37 s (SD 0.08) and 0.29 s (SD 0.07) to the probe trials, low, medium, and high tactile intensity, respectively. Long average latencies were present at the weakest tactile stimulus while intermediate durations were present in probe trials and for medium tactile intensity. A significant difference was only found between probe trials and the strongest tactile stimulus (Fig 5B, one-way ANOVA, $F(3,96) = [13.55]$, $P < 1.92 \times 10^{-7}$, Bonferroni post hoc test, $P = 0.0019$). No significant difference was found between lick latencies between probe trials and low and medium intensities.

Overall, these results demonstrate that mice were able to selectively respond to vibrations applied to their hindlimb, with responses that were not due to the acoustic noise associated to the vibration, and with a performance that scaled with the intensity of vibratory stimuli.

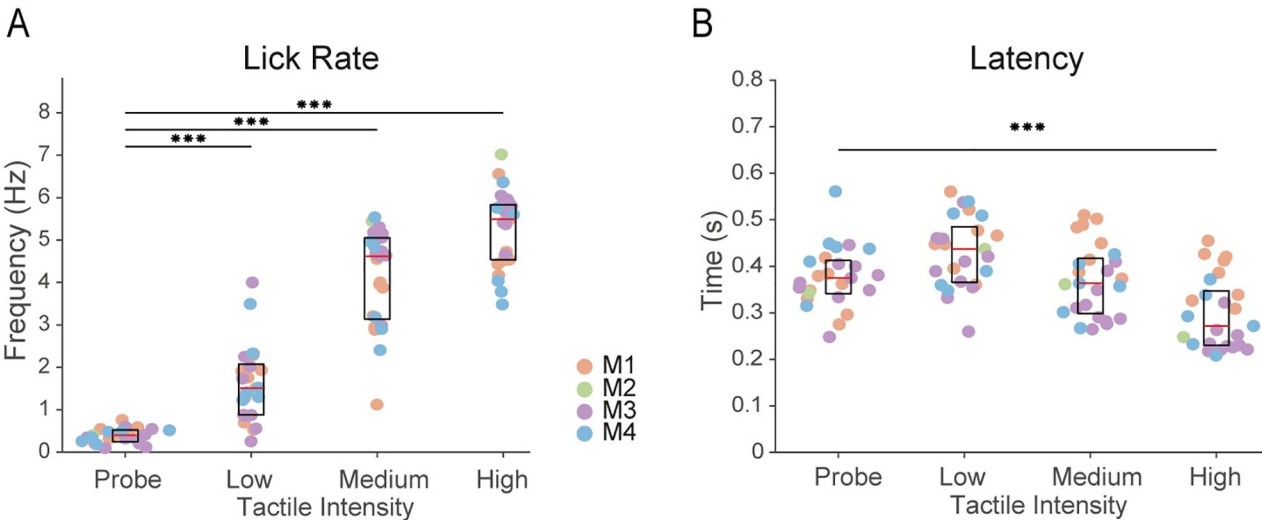

**Fig 5. Lick quantification.** (A-B) Lick rate (A) and latency (B) from all sessions across conditions. Significant differences between lick rates were found between the null condition and all other tactile intensities (***One-way ANOVA, $F$ (3, 96) = [171.4], $P$ = 2.04 x $10^{-38}$, Bonferroni post hoc test: probe vs. low intensity, $P$ = 1.09 x $10^{-5}$; probe vs. medium intensity: $P$ = 4.35 x $10^{-27}$, probe vs. high intensity, $P$ = 1.81 x $10^{-35}$). The average latencies were significantly different between probe trials and the strongest intensity (***One-way ANOVA, $F$ (3,96) = [13.55], $P < 1.92 x 10^{-7}$, Bonferroni post hoc test, $P$ = 0.0019). In all panels: Box plots show the median value with a red horizontal line; horizontal black lines indicate the 75 and 25 quartiles.

## Discussion

In this paper, we presented a novel tactile behavioural task for mice, testing their ability to detect a vibration applied to their hindlimb. We showed that mice can perform the task in a consistent way, with a performance that improved along with increasing vibratory intensities. Importantly, mice selectively responded to tactile vibrations and not to the sound produced by the vibratory motors. Evaluation of behavioural performance showed results in line with performance reported in previous rodent studies that developed a Go/No-go tactile detection task based on the whisker system [11, 55–57]. Thus, our task will allow to expand the understanding of circuit-level architecture of cortical sensory processing in the mouse hindlimb, which may enhance the translational potential of the mouse as a model for tactile processing.

### Evaluation of training regime and behavioural performance

We developed a procedure to train mice in a detection task in a limited amount of time (average of 55 days, excluding weekends). The head-fixed nature of the task makes the task suitable to be integrated with other experimental techniques such as electrophysiology (patch clamp, laminar silicone probes, tetrode recordings [58–60]), two-photon imaging [61, 62], optogenetics [63–66] and virtual reality [67] among others.

All four mice showed comparable learning curves across Stage 1 and 2. Two (M1 and M3), continued with comparable learning curves across Stage 3 and the final Stage. One (M2) displayed a fast-learning curve during Stage 3 and the final Stage. Finally, one mouse (M4) was the slowest learner but could still reach the Final Stage.

Importantly, there were no significant individual differences across mice in the psychometric parameters of performance. This indicates that the training procedure can be used by researchers to reliably achieve consistent behavioural performance.

Along Stage 3 and the final Stage, control vibrations were introduced along probe trials as a way of measuring responses to the vibratory motor sounds. Lick responses during probe trials

were considered FAs. These were lower than Hit rates for all stimulus intensities (Figs 2 and 4). Compared to Stage 2, Stage 3 took longer for animals to reach the next stage. While training time for Stage 3 was as high as 60 days, this still allowed the Final Stage (and ideally any follow-up experiment) to be performed in mice a few months old, i.e. in line with most experimental frameworks in the field. Nevertheless, the relatively long time spent in Stage 3 remains surprising, especially because mice were able to learn to perform tactile detection (Stage 2) more quickly. Interestingly, this was because, while Hit rates remained very high across Stages 2 and 3, FA rates decreased slowly and showed relatively strong fluctuations. This may be due to either a high level of impulsivity in the mouse cohort we employed or the fact that animals had difficulty learning not to respond to control vibrations during probe trials. If this was the case, it would make a strong case to always control for auditory confounds in tactile tasks.

At the end of training, mice were anyway able to selectively respond to tactile vibrations while ignoring the noise produced by the vibratory motors. Ruling out this potential confound is important for future studies with this task. Earlier studies on vibrotactile detection in mice applied a white noise stimulus to mask the sound produced by vibratory motors, but could not fully rule out that auditory information was used by mice, in addition to the vibratory frequency [33, 34]. Although we cannot fully exclude that auditory information contributed to some extent to task performance, we deem this unlikely. In fact, the relative position of the two vibratory motors was lower than the minimal discrimination angle reported in mice [41] (S1 Fig). Therefore, even if mice could detect some differences in sound volume or number of sources between real and probe trials, this would be more difficult than relying on tactile detection. Of relevance, three out of four mice quickly learned to reliably ignore the control vibration (Fig 2), and this would likely not have been the case had mice used auditory information to discriminate the two very close sources. Additional control experiments in which the position or the intensity of the second vibratory motor (not attached to the limb) are changed are however required to completely rule out the possibility that auditory information even partially contributed to task performance. Overall, the task that we introduce here successfully introduces an experimental approach to test vibrotactile detection in limbs while limiting potential auditory confounds.

All mice showed a detection performance that increased as a function of stimulus intensity. D-prime values were always higher than 1.5 at medium and high intensity vibrations. These results are comparable with previous studies with a focus on the whisker system, for example: Schwarz and Stüttgen [55] used a simple detection task in rats based on whisker deflection. They observed similar psychometric curves as a function of distinct deflection velocities (cf. Fig 1D, [55]). In addition, multiple studies in mice—see e.g. [11, 56, 57]–have shown a performance of Hit rates over 80% and FA rates lower than 30% in whisker detection tasks.

All mice showed increasing hit rates as a function of stimulus intensity. Increasing stimulus intensity was also accompanied by a proportional increase in the frequency of lick responses. This suggests that mice were not only better able to detect stimuli which had a higher vibrotactile intensity, but were also more confident in their choices, in line with previous studies indicating that lick persistence correlates with perceptual confidence [68]. Importantly, similar lick frequency values during probe trials (0.38 Hz) were found in previous studies in conditions in which animals showed very low motivation [54]. This is a further confirmation that mice were not responding to the sound generated by the vibrational motors.

Lick latencies also varied as a function of stimulus intensity (with the shortest latencies for high intensity stimuli). Curiously, probe trials presented similar latencies as the medium tactile stimulus. According to previous literature [54], lick latencies result from a combination of impulsivity [69] and the ability to inhibit a response [70]. We hypothesize that, assuming that lick latencies during probe trials reflect the baseline lick rate, only a portion of the rewarded

trials in the low-intensity condition reflect true hits, in spite of the significant D-prime values that we observed.

## Future developments

The vibrotactile detection task that we presented here can be adapted to evaluate more complex cognitive functions. In the present results, mice were set in front of one lick spout for the purposes of a Go/No-go detection task. However, this setting can be adapted into a Go/No-go discrimination task [38], or a two alternative choice task having two lick spouts instead of one—see e.g. [8, 37, 71].

The practicality of the head and leg fixation protocol allows researchers to use it in combination to the other sensory stimulations such as auditory, visual or odour stimuli [37, 59, 72–74], and also in a virtual reality setting—see e.g. [67]. Furthermore, in comparison with previous protocols developed to study vibrotactile discrimination, the mouse does not have to respond by using the same paw which is stimulated [33, 34]. Therefore, our task allows to test somatosensory processing independently from motor output, an important aspect in view of the tight connectivity between corresponding sensory and motor areas [6, 11, 57]. In future implementations of the task, it will be relevant to add pupil and snout video monitoring as well as monitor locomotory activity. Sensory stimuli are known to induce, on top of responses in thalamocortical sensory pathways, specific arousal and stereotypical motor reactions [75–78]. Movements (either spontaneous or following sensory stimuli) induce corollary discharges that evoke activity in the whole dorsal cortex (including primary sensory areas) [76, 77]. At the same time, arousal and locomotion influence sensory processing [75, 79, 80]. For all these reasons, monitoring all these variables when implementing this task or variations thereof is strongly recommended. Another aspect to consider is that some tasks have been shown not to strictly require primary sensory cortices. For instance, transient inactivation of primary somatosensory cortex impairs the detection of whisker deflections [11]. However, detection capabilities recovered about one day following permanent lesions of barrel cortex in animals previously trained for the task [56]. Thus, future studies must carefully consider to what extent the primary somatosensory cortex is required to learn and perform limb-related vibrotactile detection. Nevertheless, this task may also be useful for investigating subcortical pathways for tactile detection.

Finally, our task can be expanded and modified by modulating other parameters besides vibrational intensity (e.g. frequency and temporal patterns), but also by using other types of tactile stimuli such as texture, spatial localization and motion across the skin/fur [18].

## Conclusions

In this study we introduced a novel paradigm to investigate vibrotactile detection in mice outside of the whisker system. This paradigm is suitable to be integrated with simultaneous neural measurements and interventions, such as electrophysiology, functional imaging and optogenetics. This will enable to implement experimental paradigms in mice, to study the microcircuit-level mechanisms of tactile perception in a model with a high translational potential.

## Supporting information

**S1 Fig. Detailed overview of the experimental setup. (A)** Photo showing a top view of the experimental setup, with highlighted the positions of the vibratory motors and of the rigid plastic rings to deliver the vibration. **(B)** Schematic diagram of the experimental setup indicating the relative position of the mouse and of the vibratory motors. Positions are calculated relative to the midline between the ears. Reprinted from Biorender under a CC BY license, with

permission from Biorender, original copyright 2023.
(TIF)

## Acknowledgments

The authors would like to thank the Technology Center of the University of Amsterdam for supporting the development of the experimental set-up, and Benne Praegel, Lotte Alma and Joseph Cohen for support with behavioral training.

## Author Contributions

**Conceptualization:** Mariel Muller, Conrado A. Bosman, Umberto Olcese.

**Data curation:** Mariel Muller.

**Formal analysis:** Mariel Muller.

**Funding acquisition:** Cyriel M. A. Pennartz, Conrado A. Bosman, Umberto Olcese.

**Investigation:** Mariel Muller.

**Methodology:** Conrado A. Bosman, Umberto Olcese.

**Supervision:** Conrado A. Bosman, Umberto Olcese.

**Visualization:** Mariel Muller.

**Writing – original draft:** Mariel Muller, Umberto Olcese.

**Writing – review & editing:** Mariel Muller, Cyriel M. A. Pennartz, Conrado A. Bosman, Umberto Olcese.

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
