## [Decision Letter · Decision Letter 0]

26 Jan 2023

PONE-D-22-33136A novel task to investigate vibrotactile detection in micePLOS ONE

Dear Dr. Olcese,

Thank you for submitting your manuscript to PLOS ONE. After careful consideration, we feel that it has merit but does not fully meet PLOS ONE’s publication criteria as it currently stands. Therefore, we invite you to submit a revised version of the manuscript that addresses the points raised during the review process.

Two experts in the field have carefully reviewed the manuscript entitled, "A novel task to investigate vibrotactile detection in mice". Their comments are appended below. Both reviewers are positive for publication with leaving several critical concerns which should be considered before publication. 

This Academic Editor suggests to revise the manuscript according to the critiques. I will make the decision after receipt of your replies to each comments and necessary revision. 

We look forward to receiving your revised manuscript.

Kind regards,

Manabu Sakakibara, Ph.D.

Academic Editor

PLOS ONE

Journal Requirements:

"This study was supported by this work was supported by the European Union’s Horizon 2020 Framework Program for Research and Innovation under the Specific Grant Agreement 785907 (Human Brain Project SGA2) and 945539 (Human Brain Project SGA3) to CAB, UO and CMAP, by the FLAG-ERA JTC 2019 project DOMINO (co-financed by NWO) to CAB and UO, and by an Amsterdam Neuroscience Proof of Concept grant to UO."

6. We note that Figure 1 in your submission contain copyrighted images. All PLOS content is published under the Creative Commons Attribution License (CC BY 4.0), which means that the manuscript, images, and Supporting Information files will be freely available online, and any third party is permitted to access, download, copy, distribute, and use these materials in any way, even commercially, with proper attribution. For more information, see our copyright guidelines: http://journals.plos.org/plosone/s/licenses-and-copyright.

Reviewers' comments:

Reviewer's Responses to Questions

**Comments to the Author**

1. Is the manuscript technically sound, and do the data support the conclusions?

Reviewer #1: Partly

Reviewer #2: Yes

2. Has the statistical analysis been performed appropriately and rigorously? 

Reviewer #1: Yes

Reviewer #2: Yes

3. Have the authors made all data underlying the findings in their manuscript fully available?

Reviewer #1: Yes

Reviewer #2: Yes

4. Is the manuscript presented in an intelligible fashion and written in standard English?

Reviewer #1: Yes

Reviewer #2: Yes

5. Review Comments to the Author

Reviewer #1: In this manuscript, Muller and colleagues propose a go-no go task based on vibrotactile stimuli on mice. I think the protocol is well thought out and designed and could be used to infer the neuronal mechanism of somatosensation.

In the introduction section, the authors propose their protocol as a better alternative to the study of somatosensation compared to studies involving whiskers. I think this approach simply represents an alternative, not necessarily the best. Much of the sensory cortex of mice is barrel-dominated. Moreover, a strong specificity between individual whiskers and groups of neurons responding to stimulation is present. This makes it possible to specifically analyse groups of neurons that respond to the stimulation of a single whisker.

I think this is a flaw in form rather than content. Indeed, in the discussion session the authors are more cautious in describing the translational potential of their protocol. I understand that the authors need to show their work in an appealing light, but I would suggest that the authors rewrite the introduction giving credit to previous studies that analysed sensory responses from whiskers and propose their method as an alternative, which is equally valid.

The main point of clarification for me is the use of PV-Cre mice for this study. The authors should thoroughly clarify why they used this model in spite of a non-transgenic line. There is evidence in the literature, albeit conflicting, showing that being PV expressed in sperm, it is possible that an unintended percentage of germline or global recombination may occur. This is particularly true for Pvalb-2A-Cre (PMID: 24137112). Neurons expressing parvalbumin are crucial for the correct processing of sensory stimuli in the cortex, and it is also present in the peripheral nervous system, e.g., in the dorsal root ganglia (DRG).

I ask the authors to clarify why they used this model and to specify whether their Pvalb-Cre model is Pvalb-IRES-Cre (Hippenmeyer et al., 2005) or Pvalb-2A-Cre (Madisen et al., 2010). If it is the former, this could be a major concern for the results and implications of this study.

Minor points:

I would add a photo of the experimental setup, this could help, more than the diagram, to replicate the work. This could help in understanding how the vibrating motor was applied.

Please, specify the frequency of vibration stimuli applied in each of the stages.

The figures of the manuscript appear of a low quality, please check and revise accordingly.

Reviewer #2: In this study, the authors develop a new behavioral paradigm wherein mice detect tactile sensory stimuli to get reward. They find that mice learn the task in two months of training and report fine details of their training processes. This task will be a good addition to the field working on sensorimotor transformation and allow future investigations to test how similar/dissimilar underlying circuitries are between different sensory modalities. Also, the manuscript is clearly written. I only have one major comment and quite a few minor points need to be addressed.

Major

1) The authors argue that mice are not using auditory stimulus to make decisions by introducing the control vibration stimulus. However, it is still possible that mice use sounds at least in part in the current task design that lacks some control experiments. First, given that the control vibration is given in both Go and No-Go trials, mice could discriminate the sound volume between the two trial types unless the control vibration have several different intensities in both trial types. Second, mice might discriminate the location and/or the number of the sound sources. If mice use auditory information, the current task design would be very confusing for mice since identical sound (i.e. the control stimulation) gives reward at 50% probability. Possibly this is one of the reasons why it takes too long for mice to get this very simple sensory detection task, which usually takes only a few days in other modalities. The authors mention that the two sound sources are close with each other, but no details are provided (e.g. how much degrees are they separated in relation to the left ear? How much spatial resolution does the mouse auditory system have?). One easy solution could be swapping or randomizing the sound locations within a session. This should be discussed as a potential caveat.

Minor

General comments

2) Another concern is if mice are using ‘specific’ sensory information as the authors aimed for or mice just have high arousal in response to vibration stimulus which can trigger higher lick probability. Adding pupil dilation data would be highly recommended if they have. Otherwise, this should be discussed.

3) Since simple visual detection task doesn’t require the primary visual cortex in mice (e.g. Resulaj et al., 2018), it is possible that this task doesn’t involve S1. In this regard, it would be informative if the authors mention how similar/dissimilar the subcortical circuitries are between mice and humans as well.

4) As mentioned above, the task takes surprisingly long time, but not “with reasonably short training times” at all. The authors should discuss why it takes long or how it can be shortened. This kind of information would be useful for the field.

Specific comments

5) Why are PV-Cre lines used?

6) Add references to line 48 and 76.

7) line 99: 3% 'for' induction and 1.5% 'for' maintenance. The manuscript is overall well written but there are several incorrect English in grammar. Authors should go through the manuscript more carefully.

8) I doubt if the mouse in Figure 1B really got the task since there are significant amount of licking prior to sensory stimuli – in several trials, the lick rate looks over 10 Hz. The author should be aware that the major increase in licking rate in response to stimulus comes mainly from consumption of the reward. My concern here is that this mouse is just making spontaneous licking which triggers valve opening if it happens in the right timings. Supporting this, the licking does not seem to be well time-locked relative to stimulus onset. The data should be replaced with that of expert mouse. The current one is just confusing.

9) The authors provide fine details of procedures in each training stage. It would be nicer if they explain the rationales or aims of each stage more explicitly.

10) Line 198: Can the performance really be assessed by how much milk was consumed given that reward was given automatically in this stage? I guess reaction time would be a better measure here.

11) Why it takes 13 days in the first stage even though it just gives automatic rewards? Better to describe a bit more about mouse behavior. Are mice just freaked out in the setup?

12) How can 5-20 sec of ITI have an exponential distribution? I suspect they set cut-off values on both upper and lower ends but cannot find any information. What is the lambda? Does it still have flat hazard rate after thresholding?

13) Line 231: What is the rationale of taking 1 sec window? Given that stimulation is given for 2 sec, shouldn’t it be 2 sec?

14) Line 232: D-prime is not defined yet in the manuscript. Also, I don’t see the point of calculating D-prime without FA rate. Better to use different measures or define it with a different equation in the same paragraph. Also, Figure 2D is almost meaningless in the current way since different measures are used in the same graph.

15) Figure 2C should show all four mice.

16) Line 261. If Hit and FA rates are rounded to 1, how can the normal inverse cumulative distribution function be calculated? People do the other way around (see Stanislaw & Todorov, 1999).

17) Using different words for the same thing is very confusing. Probe trials vs No-Go trials; and control vibration vs null stimulus. Use only one of them.

18) In Figure 3A, why does the mouse show more spontaneous licking prior to stimulus onset in No-Go trials than Go trials? Are Go and No-Go trials randomly given? Or did you put any bias depending on performance in the previous trials?

19) Figure 4A: how can d-prime calculated for Null? This is very confusing if you use the equation in line 259.

20) Figure 4C: how can Hits be counted for Null? Y-axis should be corrected accordingly. Also, what does “E.g” stand for?

21) Figure 4F and 4G: Just show hit and FA rates, which describe mouse behavior better. Current measures in F and G are values obtained after curve fitting on small number of data points, which has less meaning. It is hard to get how the values in 4e were calculated from the manuscript, but the same logic can be true for 4E since the steepness of curves would be very different from the reality, depending on where each mouse put its threshold.

22) Figure 5A: this figure panel has almost no information. Again, the lick rate in Go trials mainly represent how vigorously mice are licking a spout during reward consumption. Also, how lick rate is calculated is unclear. Is it coming from only hit trials? Only during the 1 sec of stimulation period was considered (if so, it is redundant to 5B)? Reaction time (5B) is enough here and lick rate is just confusing.

23) Figure 5B: show non-significant comparisons as well (in a similar way as 5A).

24) line 112: what does PVC stand for?

6. PLOS authors have the option to publish the peer review history of their article (what does this mean?). If published, this will include your full peer review and any attached files.

Reviewer #1: No

Reviewer #2: No

---

## [Author Response · Author response to Decision Letter 0]

26 Mar 2023

Responses to the reviewers’ comments

Reviewer #1: In this manuscript, Muller and colleagues propose a go-no go task based on vibrotactile stimuli on mice. I think the protocol is well thought out and designed and could be used to infer the neuronal mechanism of somatosensation.

We thank the reviewer for the positive assessment of our manuscript. Below we have addressed all the points that were raised.

In the introduction section, the authors propose their protocol as a better alternative to the study of somatosensation compared to studies involving whiskers. I think this approach simply represents an alternative, not necessarily the best. Much of the sensory cortex of mice is barrel-dominated. Moreover, a strong specificity between individual whiskers and groups of neurons responding to stimulation is present. This makes it possible to specifically analyse groups of neurons that respond to the stimulation of a single whisker.

I think this is a flaw in form rather than content. Indeed, in the discussion session the authors are more cautious in describing the translational potential of their protocol. I understand that the authors need to show their work in an appealing light, but I would suggest that the authors rewrite the introduction giving credit to previous studies that analysed sensory responses from whiskers and propose their method as an alternative, which is equally valid.

We concur with the reviewer about the need to adjust the tone of the introduction (and also of the abstract). We have now addressed this by reformulating the relevant paragraphs.

First, the last sentence of the abstract has been modified as follows:

“Thus, this study introduces a novel task to study the neuron-level mechanisms of tactile processing in a system other than the more commonly studied whisker system.”

The introduction has then been changed by adding the following sentence:

“In rodents, the barrel field occupies a large fraction of the somatosensory cortex, and, thanks to the highly specific mapping between individual whiskers and topographically organized groups of neurons, this model system has allowed to understand many properties of tactile processing, ranging from stimulus selectivity [6,20] to experience-dependent plasticity [21–23].”

We have also modified the last paragraph of the Introduction to make it clear that we only aim to provide an alternative tool:

“To address this, we have developed a novel vibrotactile detection task for head-fixed mice that provides an alternative framework to those available for the barrel/whisker system to study the detection of tactile stimuli as a function of the intensity of the applied stimulus. [...] By showing that mice can successfully be trained to perform this task, we have provided an alternative tool that neuroscientists can use to investigate the neuron-level mechanisms of touch perception and decision-making in a system other than the barrel/whisker cortex.”

The main point of clarification for me is the use of PV-Cre mice for this study. The authors should thoroughly clarify why they used this model in spite of a non-transgenic line. There is evidence in the literature, albeit conflicting, showing that being PV expressed in sperm, it is possible that an unintended percentage of germline or global recombination may occur. This is particularly true for Pvalb-2A-Cre (PMID: 24137112). Neurons expressing parvalbumin are crucial for the correct processing of sensory stimuli in the cortex, and it is also present in the peripheral nervous system, e.g., in the dorsal root ganglia (DRG).

I ask the authors to clarify why they used this model and to specify whether their Pvalb-Cre model is Pvalb-IRES-Cre (Hippenmeyer et al., 2005) or Pvalb-2A-Cre (Madisen et al., 2010). If it is the former, this could be a major concern for the results and implications of this study.

We apologize for not specifying this important aspect better. We have now indicated that the transgenic line we employed is the Pvalb-IRES-Cre line, thus reducing the risks arising from possible recombination. The motivation for choosing this line is that it is commonly employed (alongside other lines in which Cre is expressed in all GABAergic neurons) to implement optogenetic-based inactivation of specific areas. Furthermore, in our experience, we have used this line for complex sensory-processing tasks, in which satisfactory performance was obtained, comparable with that of wild-type C57BL6 mice. For these reasons, we believe that the use of a transgenic line does not influence the results we present. To address this point, we have added the following sentences to the Methods & Results sections:

“This transgenic line is often used in studies in which area-specific optogenetic inactivation is used, and has been successfully used as an experimental model to study sensory processing and decision making – see e.g. [8,34,37]. Comparable results are therefore expected with different mouse lines.”

Minor points:

I would add a photo of the experimental setup, this could help, more than the diagram, to replicate the work. This could help in understanding how the vibrating motor was applied.

We have now included an annotated photo of the experimental setup as S1A Fig. However, since the paws are occluded in view of the fact that mice were placed in a plastic tube (to make them comfortable), the exact positioning of the vibratory motors can be hard to see. For this reason, we would rather keep a schematic depiction in Fig 1.

Please, specify the frequency of vibration stimuli applied in each of the stages.

We have now provided information about the frequency and intensity of vibration applied in each stage. First of all, we have now indicated the precise type of vibratory motor that we used:

“The plastic ring was attached to a vibratory motor (LilyPad- SparkFun-DEV11008, where the motor is a 310-101 10mm shaftless vibration motor from Precision Microdrive)”

We have then indicated frequency and intensity of the vibration in both the training stages and the final stage:

“An Arduino board (Mega 2560) was used to control the intensity and duration of the vibration. Intensity could be modulated on a 0-255 pulse width modulation (PWM) scale level. In the training stages, PWM scale value was set at the maximum value, corresponding to an intensity of 3.6-3.8 G with a vibration frequency of 210-220 Hz.”

“Specifically, we used three intensity ranges that matched to a below-chance response (<50%, low intensity: 1.8-2.25 G at a frequency of 105-130 Hz), a higher than chance but moderate response (50-80%, medium intensity: 2.6-2.9 G at a frequency of 150-175 Hz), or a higher than 80% hit rate (high intensity: 3.3-3.8 G at a frequency of 195-210 Hz).”

To account for the fact that the vibratory motors based on the principle of eccentric rotating mass vary both rotation speed and intensity as a function of input voltage, we also added the following discussion point:

“Of relevance, mice did not need additional training sessions to habituate to the novel stimulus intensities. This suggests that the different vibratory frequency that the type of vibratory motors we used produce as a function of applied input voltage and vibratory intensity was either not perceived by mice or did not affect their detection ability, which was rather mainly influenced by intensity.”

The figures of the manuscript appear of a low quality, please check and revise accordingly.

We have uploaded high-resolution files for each figure. These need to be downloaded from the submission system, as only lower resolution versions are included in the pdf for revision.

Reviewer #2: In this study, the authors develop a new behavioral paradigm wherein mice detect tactile sensory stimuli to get reward. They find that mice learn the task in two months of training and report fine details of their training processes. This task will be a good addition to the field working on sensorimotor transformation and allow future investigations to test how similar/dissimilar underlying circuitries are between different sensory modalities. Also, the manuscript is clearly written. I only have one major comment and quite a few minor points need to be addressed.

We thank the reviewer for the very careful assessment of our manuscript and for spotting all these issues that, albeit mostly minor, impacted its quality.

Major

1) The authors argue that mice are not using auditory stimulus to make decisions by introducing the control vibration stimulus. However, it is still possible that mice use sounds at least in part in the current task design that lacks some control experiments. First, given that the control vibration is given in both Go and No-Go trials, mice could discriminate the sound volume between the two trial types unless the control vibration have several different intensities in both trial types. Second, mice might discriminate the location and/or the number of the sound sources. If mice use auditory information, the current task design would be very confusing for mice since identical sound (i.e. the control stimulation) gives reward at 50% probability. Possibly this is one of the reasons why it takes too long for mice to get this very simple sensory detection task, which usually takes only a few days in other modalities. The authors mention that the two sound sources are close with each other, but no details are provided (e.g. how much degrees are they separated in relation to the left ear? How much spatial resolution does the mouse auditory system have?). One easy solution could be swapping or randomizing the sound locations within a session. This should be discussed as a potential caveat.

We agree with the reviewer about the need to provide additional information and discussions concerning potential confounding factors. 

The reviewer is correct in pointing out that mice could, in theory, discriminate go/no-go trials in terms of overall auditory volume or a number of other sound sources. However, this would arguably be more difficult than relying on tactile information, especially in view of the fact that the location of the two vibratory motors was very close (~3 cm) and at a close angle (~ 10 deg, see S1B Fig) with respect to the ears. Directional acuity in mice has been reported to have a minimum audible angle of 31 deg (Lauer et al., 2011). In our setup, therefore, the two vibratory motors are closer than the minimum angle required for two separate sources to be discriminated. For what pertains to the relatively long time required by mice to learn the task, this was compatible with other Go/NoGo tasks that we have implemented in the past, at least for what pertains to the initial stages (in which the additional vibration was not added). Except for one mouse (M4), all other mice relatively quickly learned to ignore the additional vibration (see Fig 2B, stage 3). In fact, the most time-consuming component of training was reducing the false alarm rate in stage 2 (Fig 2C). In conclusion, while we believe that using tactile information is most likely, we cannot exclude that auditory information could at least partially have contributed to task performance. The control experiments suggested by the reviewer could at least partially address this issue, and we agree that they need to be discussed.

To address the reviewer’s concerns, we have updated the Methods & Results section with additional details about the location of the secondary vibratory motor, and we have displayed this in S1B Fig:

“[…] the angle between the two motors was set at about 10 deg (see S1B Fig). This is lower than the minimal auditory angle of 31 deg that has been reported in mice [41].”

Furthermore, we have added the following paragraph to the Discussion section:

“Although we cannot fully exclude that auditory information contributed to some extent to task performance, we deem this unlikely. In fact, the relative position of the two vibratory motors was lower than the minimal discrimination angle reported in mice [41] (S1 Fig). Therefore, even if mice could detect some differences in sound volume or number of sources between real and probe trials, this would be more difficult than relying on tactile detection. Of relevance, three out of four mice quickly learned to reliably ignore the control vibration (Fig 2), and this would likely not have been the case had mice used auditory information to discriminate the two very close sources. Additional control experiments in which the position or the intensity of the second vibratory motor (not attached to the limb) are changed are, however, required to completely rule out the possibility that auditory information even partially contributed to task performance.”

Minor

General comments

2) Another concern is if mice are using ‘specific’ sensory information as the authors aimed for or mice just have high arousal in response to vibration stimulus which can trigger higher lick probability. Adding pupil dilation data would be highly recommended if they have. Otherwise, this should be discussed.

Unfortunately, we do not have video recordings of the animals’ eyes. We fully agree that it would have been relevant information. We have added a note about this in the Discussion section:

“In future implementations of the task, it will be relevant to add pupil and snout video monitoring as well as monitor locomotory activity. Sensory stimuli are known to induce, on top of responses in thalamocortical sensory pathways, specific arousal and stereotypical motor reactions [75–78]. Movements (either spontaneous or following sensory stimuli) induce corollary discharges that evoke activity in the whole dorsal cortex (including primary sensory areas) [76,77]. A the same time, arousal and locomotion influence sensory processing [75,79,80]. For all these reasons, monitoring all these variables when implementing this task or variations thereof is strongly recommended.

3) Since simple visual detection task doesn’t require the primary visual cortex in mice (e.g. Resulaj et al., 2018), it is possible that this task doesn’t involve S1. In this regard, it would be informative if the authors mention how similar/dissimilar the subcortical circuitries are between mice and humans as well.

We agree with the reviewer that this is an interesting point. Previous studies in the whisker system show intriguing results about the role of cortex in stimulus detection. For instance, the study by Sachidhanandam et al. (2013) showed how the barrel cortex was necessary for detecting whisker deflections. However, following permanent lesions, mice are able to recover behavioral performance already one day after the lesion (Hong et al., 2018). Nevertheless, this later study also established that S1 is necessary to learn a tactile detection task. Thus, we can expect a similar role of somatosensory cortex in our task. Concerning other sensory modalities, to our knowledge, V1 has been found to be necessary for visual detection. This includes the study by Resulaj et al. (2018), which showed that the first 40-80 ms of visually-evoked activity in V1 are required for visual discrimination. Other studies have in fact shown that some forms of visually-guided behavior do not require V1 but mostly for innate simple detection tasks. Nonetheless, visual performance is in any case degraded following V1 inactivation (see e.g, Glickfeld et al., 2013). Overall, these results point towards a similarity between cortical systems devoted to processing distinct modalities, in which the visual cortex is generally necessary to detect visual stimuli. Following permanent inactivation, however, the performance of a previously learned task can be recovered via compensation through subcortical mechanisms. Nevertheless, while sensory-motor transformations might sometimes be performed without the cortex, perception itself is thought to require cortical processing, in the absence of which a subject would experience a form of blindsight (Cowey and Stoerig, 1995). Irrespective of all these considerations, we believe that the task that we developed can be a useful tool to study tactile processing, even in subcortical systems. However, we think an extensive discussion of the similarities of subcortical somatosensory processing pathways between rodents and humans is beyond the scope of our manuscript. To address the reviewer’s comments, we have thus added a paragraph to the Discussion section:

“Another aspect to consider is that some tasks have been shown not to strictly require primary sensory cortices. For instance, transient inactivation of primary somatosensory cortex impairs the detection of whisker deflections [11]. However, detection capabilities recovered about one day following permanent lesions of barrel cortex in animals previously trained for the task [56]. Thus, future studies must carefully consider to what extent the primary somatosensory cortex is required to learn and perform limb-related vibrotactile detection. Nevertheless, this task may also be useful for investigating subcortical pathways for tactile detection.”

4) As mentioned above, the task takes surprisingly long time, but not “with reasonably short training times” at all. The authors should discuss why it takes long or how it can be shortened. This kind of information would be useful for the field.

We agree that this is an important aspect. As shown in Fig 2, mice have no difficulty achieving high Hit rates. Rather, it is achieving consistently low FA rates that require time. Given the longer times mice spend in Stage 3 compared to Stage 2, this may be due to the difficulty in discriminating auditory stimuli (due to the control vibration) from purely tactile stimuli. We have now added an additional paragraph discussing training time:

“Compared to Stage 2, Stage 3 took longer for animals to reach the next stage. While training time for Stage 3 was as high as 60 days, this still allowed the Final Stage (and ideally any follow-up experiment) to be performed in mice a few months old, i.e. in line with most experimental frameworks in the field. Nevertheless, the relatively long time spent in Stage 3 remains surprising, especially because mice were able to learn to perform tactile detection (Stage 2) more quickly. Interestingly, this was because, while Hit rates remained very high across Stages 2 and 3, FA rates decreased slowly and showed relatively strong fluctuations. This may be due to either a high level of impulsivity in the mouse cohort we employed or the fact that animals had difficulty learning not to respond to control vibrations during probe trials. If this was the case, it would make a strong case to always control for auditory confounds in tactile tasks.”

In view of this, we would like to maintain the phrasing “reasonably short training times” in the abstract.

Specific comments

5) Why are PV-Cre lines used?

The motivation for choosing this line is that it is commonly employed (alongside other lines in which Cre is expressed in all GABAergic neurons) to implement optogenetic-based inactivation of specific areas. Furthermore, in our experience, we have used this line for complex sensory-processing tasks, in which satisfactory performance was obtained, comparable with that of wild type C57BL6 mice. For these reasons, we believe that using a transgenic line does not influence the results we present. To address this point, we have added the following sentences to the Methods & Results sections:

“This transgenic line is often used in studies in which area-specific optogenetic inactivation is used, and has been successfully used as an experimental model to study sensory processing and decision making – see e.g. [8,34,37]. Comparable results are therefore expected with different mouse lines.”

6) Add references to line 48 and 76.

We have added references to these lines.

7) line 99: 3% 'for' induction and 1.5% 'for' maintenance. The manuscript is overall well written but there are several incorrect English in grammar. Authors should go through the manuscript more carefully.

We have now carefully checked the whole manuscript and made several adjustments.

8) I doubt if the mouse in Figure 1B really got the task since there are significant amount of licking prior to sensory stimuli – in several trials, the lick rate looks over 10 Hz. The author should be aware that the major increase in licking rate in response to stimulus comes mainly from consumption of the reward. My concern here is that this mouse is just making spontaneous licking which triggers valve opening if it happens in the right timings. Supporting this, the licking does not seem to be well time-locked relative to stimulus onset. The data should be replaced with that of expert mouse. The current one is just confusing.

We agree with the reviewers that it is better to show a licking raster plot from an expert mouse, rather than from an example trial from Stage 3. We have replaced the panel and adjusted the figure legend.

9) The authors provide fine details of procedures in each training stage. It would be nicer if they explain the rationales or aims of each stage more explicitly.

We thank the reviewer for this advice. We have now included the following additional explanations at the beginning of the sections explaining each stage:

Stage 1: 

“The goal of this stage was to teach animals to associate vibrotactile stimuli to the delivery of a reward.”

Stage 2:

“Thus, after associating stimulus presentation to reward delivery (Stage 1), in Stage 2 mice had to learn to perform licking actions to obtain a reward, and to time licking actions to sensory stimuli.”

Stage 3:

“The goal of Stage 3 was to train animals to respond to vibrotactile stimuli while ignoring sounds produced by the motors.”

Final Stage:

“The goal of the Final Stage was to be able to obtain psychometric curves quantifying task performance.”

10) Line 198: Can the performance really be assessed by how much milk was consumed given that reward was given automatically in this stage? I guess reaction time would be a better measure here.

We agree with the reviewer that in theory reaction times could be used in this stage. However, reward delivery can sometimes induce an artefact (false positive) in the lick detection system. Since this artefact cannot be univocally distinguished from a licking action, we chose to use an indirect measure to quantify progress within Stage 1. Note that this is not a problem for evaluating reaction times in active stages (Stages 2 and following), since reward delivery occurs after the detection of the first lick. Furthermore, we were able to remove this artefact during the course of Stage 1. In Stage 1 the amount of milk consumed during each session is thus a good indicator of how much animals are able to associate the reward spouts to milk delivery. What we observed is that, after a few days (2-3, on average), mice were able to consume a large fraction of the provided milk (1.2-1.5 ml). More details are provided in our answer to the next comment. We have now added a sentence to acknowledge that reaction times could be used in lieu of the amount of consumed reward:

“Of relevance, reaction times could potentially also be used to monitor mouse behavior during Stage 1.”

11) Why it takes 13 days in the first stage even though it just gives automatic rewards? Better to describe a bit more about mouse behavior. Are mice just freaked out in the setup?

We agree that 13 days is a long period of time for Stage 1. In fact, we saw that all mice could consume a large fraction of the provided milk (1.2-1.5 ml) before becoming satiated. However, we also noticed that placing the vibratory ring around the animal’s hindlimb required some habituation and some training for experimenters. While mice did not fight against the vibratory ring, we did notice some discomfort during the preparation phase. Also, experimenters needed some time to learn the procedure. For this reason, we decided to take some extra time during this phase.

All this is now better explained in the following paragraph:

“All mice learnt to lick to obtain rewards within a period of 4 days, but we decided to extend Stage 1 for up to 13 days (Fig 2B) before proceeding to Stage 2. We chose this longer habituation period because we wanted to make sure that mice were well habituated to the procedure of positioning the vibratory ring to the hindlimb. Furthermore, experimenters also needed some sessions to learn to master this step, in order to minimize discomfort to the mice and ensure a stable positioning. Stage 1 could therefore be shortened in future experiments.”

12) How can 5-20 sec of ITI have an exponential distribution? I suspect they set cut-off values on both upper and lower ends but cannot find any information. What is the lambda? Does it still have flat hazard rate after thresholding?

We apologize if this was not clear. We have now added the following additional description:

“The exponential distribution had a mean of 1s, with a minimum (offset) value of 5 s and a cutoff at 20 s.”

A minimal duration for the ITI is necessary to guarantee a minimal separation between subsequent trials and is common practice. As the reviewer correctly points out, the flat hazard rate is theoretically not respected for values higher than 20 s. However, the probability of obtaining a ITI longer than 20s is smaller than 0.001% and therefore negligible. This is discussed in an additional sentence:

“Note that the probability of having an ITI longer than 20 s is smaller than 0.001%. Thus the hazard rate can be assumed as flat for ITI values larger than 5 s.”

We also updated the description of the ITI duration for the Final Stage.

13) Line 231: What is the rationale of taking 1 sec window? Given that stimulation is given for 2 sec, shouldn’t it be 2 sec?

Here we only aimed to establish the mice’ spontaneous lick rate in a period close to stimulus delivery, to estimate the false alarm rate. Hence it is not necessary to use the same duration as the stimulation window. This is now better explained:

“‘False alarms’ (FAs) responses were estimated by considering licks detected during ITIs, in the one second window before the stimulus period (a window that was chose to assess spontaneous lick rate in the vicinity of stimulus delivery).”

14) Line 232: D-prime is not defined yet in the manuscript. Also, I don’t see the point of calculating D-prime without FA rate. Better to use different measures or define it with a different equation in the same paragraph. Also, Figure 2D is almost meaningless in the current way since different measures are used in the same graph.

We apologize for not defining D-prime at this stage. This has now been addressed by moving the definition earlier. What changes between Stage 2 and the following stages is how FAs are calculated. In fact, we do calculate FAs in Stage 2, which allows us to compute D-prime values. Specifically, in stage 2 FAs were computed by considering the 1 s window just preceding stimulus onset for trial in which ITI was at least 6 s long. These are temporal windows that match the temporal properties of regular trials and thus allow us to assess a surrogate FA rate by computing the fraction of windows in which a lick was present. A similar procedure has been used in previous studies, e.g. (Oude Lohuis et al., 2022). During Stage 3, FAs were assessed during the 1 s window within the ITI in which the vibratory control stimulus was presented. Importantly, this control vibration was introduced during the ITIs in Stage 3 and had a random onset that made it compatible with the temporal statistics of regular trials. We have now added the following paragraphs to better explain how FAs were computed:

Stage 2:

“‘False alarms’ (FAs) responses were estimated by considering licks detected during ITIs, in the one second window before the stimulus period (a window that was chosen to assess spontaneous lick rate in the vicinity of stimulus delivery and that has temporal properties comparable to those of regular trials). The FA rate was computed as the fraction of 1 s windows preceding stimulus onset that included at least one lick event, which were thus considered as surrogate FAs, or trials in which mice would have licked even in the absence of sensory stimuli [37].

Performance was measured by computing the D-prime (d’) sensitivity index [15,49], that measures the difference between Hit rate and FA rate. D-prime was computed via the following formula:

d^'=ɸ^(-1) (Hit rate)-ɸ^(-1) (FA rate) (1)

where ɸ(-1 )is the inverse of the cumulative normal distribution of the hit rate and the false alarm rate We adjusted values of Hit and FA rates that were equal to 1 or 0 to either 0.99 or 0.01, respectively [50].” 

Stage 3:

“False alarm responses were based on licks happening during control vibrations within ITIs. The absence of licks during control vibrations presented within ITIs were considered as a CR response, their presence as a FA.”

15) Figure 2C should show all four mice.

We have update Fig 2C and Fig 2D to show all four mice.

16) Line 261. If Hit and FA rates are rounded to 1, how can the normal inverse cumulative distribution function be calculated? People do the other way around (see Stanislaw & Todorov, 1999).

We apologize for the typo. We actually meant the opposite. Values of Hit and FA rates equal to 1 were updated to a value of 0.99 (and rates equal to 0 to a value of 0.01), following for instance Macmillan & Creelman (2005). We have amended the relevant sentences.

17) Using different words for the same thing is very confusing. Probe trials vs No-Go trials; and control vibration vs null stimulus. Use only one of them.

We have adjusted this, by using the term “control vibration” for Stage 3 and “probe trial” for the Final Stage. We believe that different terms are important because the control vibrations are introduced in Stage 3 during the ITI, while actual probe trials are only included in the Final Stage. We also refer to Go trials as “tactile trials”.

18) In Figure 3A, why does the mouse show more spontaneous licking prior to stimulus onset in No-Go trials than Go trials? Are Go and No-Go trials randomly given? Or did you put any bias depending on performance in the previous trials?

Tactile and probe trials were randomly given, with no trial selection based on performance in previous trials. This is now clarified in the text. For what pertains Fig 3A, we actually have no explanation for the behavior of this mouse, but this is likely to be a random occurrence. As this is not representative of other sessions, we have replaced this example with a more representative one.

19) Figure 4A: how can d-prime calculated for Null? This is very confusing if you use the equation in line 259.

We apologize for this mistake. We have remove probe trials from Fig 4A. Indeed, the D-prime was not computed for probe trials.

20) Figure 4C: how can Hits be counted for Null? Y-axis should be corrected accordingly. Also, what does “E.g” stand for?

Hits during probe trials (“Null condition”) are actually FAs. We have now changed the label of the y axis as “Response rate” to prevent any confusion.

Regarding ‘E.g.’, we agree that it may be confusing. We have replaced this label with “Ex. Fig 3”, as this better reflect what is also explained in the legend.

21) Figure 4F and 4G: Just show hit and FA rates, which describe mouse behavior better. Current measures in F and G are values obtained after curve fitting on small number of data points, which has less meaning. It is hard to get how the values in 4e were calculated from the manuscript, but the same logic can be true for 4E since the steepness of curves would be very different from the reality, depending on where each mouse put its threshold.

We agree with the reviewer that believe that the measures shown in panels 4E-G may be affected by noise. However, we also believe that these are informative, in particular to show that relatively narrow ranges of values can be observed. Moreover, even if the estimation of sensitivity, lapse rate and guess rate is affected by the relatively low number of data points, these parameters remain more informative than just presenting the psychometric curves (Fig 4C). For this reason, we would prefer to keep these panels in Fig 4.

22) Figure 5A: this figure panel has almost no information. Again, the lick rate in Go trials mainly represent how vigorously mice are licking a spout during reward consumption. Also, how lick rate is calculated is unclear. Is it coming from only hit trials? Only during the 1 sec of stimulation period was considered (if so, it is redundant to 5B)? Reaction time (5B) is enough here and lick rate is just confusing.

We apologize if this panel was not described with sufficient clarity. The lick rate was calculated as number of licks during the 1 s stimulus presentation window. We only included hit trials for conditions with tactile stimuli, and FAs for probe trials. While reaction times are important, we believe (as can be observe in Fig 5A) that lick rates offer a different kind of information, more related to perceptual confidence (see e.g. Schmack et al., Science, 2021). In fact, when comparing Figs 5A and 5B, one can see that lick rates – but not reaction times – are monotonically related to hit rates (and likely reflect perceptual confidence). We have now better described the data displayed in Fig 5A in the following paragraph:

“Lick rates were estimated during the 1 s window of stimulus presentation (or the equivalent window during probe trials). We only included hit trials and FA trials.”

We have also added a discussion point about this: 

“All mice showed increasing hit rates as a function of stimulus intensity. Increasing stimulus intensity was also accompanied by a proportional increase in the frequency of lick responses. This suggests that mice were not only better able to detect stimuli which had a higher vibrotactile intensity, but were also more confident in their choices, in line with previous studies indicating that lick persistence correlates with perceptual confidence [68]. ”

23) Figure 5B: show non-significant comparisons as well (in a similar way as 5A).

Fig 5A showed significant differences between multiple conditions, in contrast with Fig 5B, in which we found only a significant difference between probe trials and the strongest tactile intensity. Both Figures therefore only indicate significant differences.

24) line 112: what does PVC stand for?

PVC stands for polyvinyl chloride. We have now indicated this.

---

## [Decision Letter · Decision Letter 1]

10 Apr 2023

A novel task to investigate vibrotactile detection in mice

PONE-D-22-33136R1

Dear Dr. Olcese,

We’re pleased to inform you that your manuscript has been judged scientifically suitable for publication and will be formally accepted for publication once it meets all outstanding technical requirements.

Kind regards,

Manabu Sakakibara, Ph.D.

Academic Editor

PLOS ONE

Additional Editor Comments (optional):

Reviewers' comments:

Reviewer's Responses to Questions

**Comments to the Author**

1. If the authors have adequately addressed your comments raised in a previous round of review and you feel that this manuscript is now acceptable for publication, you may indicate that here to bypass the “Comments to the Author” section, enter your conflict of interest statement in the “Confidential to Editor” section, and submit your "Accept" recommendation.

Reviewer #1: All comments have been addressed

Reviewer #2: All comments have been addressed

2. Is the manuscript technically sound, and do the data support the conclusions?

Reviewer #1: Yes

Reviewer #2: Yes

3. Has the statistical analysis been performed appropriately and rigorously? 

Reviewer #1: Yes

Reviewer #2: Yes

4. Have the authors made all data underlying the findings in their manuscript fully available?

Reviewer #1: Yes

Reviewer #2: Yes

5. Is the manuscript presented in an intelligible fashion and written in standard English?

Reviewer #1: Yes

Reviewer #2: Yes

6. Review Comments to the Author

Reviewer #1: (No Response)

Reviewer #2: (No Response)

7. PLOS authors have the option to publish the peer review history of their article (what does this mean?). If published, this will include your full peer review and any attached files.

Reviewer #1: No

Reviewer #2: No

---

## [Editor Report · Acceptance letter]

12 Apr 2023

PONE-D-22-33136R1 

A novel task to investigate vibrotactile detection in mice 

Dear Dr. Olcese:

I'm pleased to inform you that your manuscript has been deemed suitable for publication in PLOS ONE. Congratulations! Your manuscript is now with our production department. 

Kind regards, 

on behalf of

Dr. Manabu Sakakibara 

Academic Editor

PLOS ONE